# Unsupervised Learning of Audio Perception:
# Learning to Project Data to UMAP space

**Prateek Verma** [1]   **Kenneth Salisbury** [1]

## Abstract

Audio perception is a key to solving a variety of problems ranging from acoustic scene analysis, music meta-data extraction, recommendation, synthesis and analysis. It can potentially also augment computers in doing tasks that humans do effortlessly in day-to-day activities. This paper builds upon key ideas to build perception of touch sounds without access to any ground-truth data. We show how we can leverage ideas from classical signal processing to get large amounts of data of any sound of interest with a high precision. These sounds are then used, along with the images to map the sounds to a clustered space of the latent representation of these images using neural architectures. The model trained to map sounds to this clustered representation, gives reasonable performance in mapping to the respective image associated with the sound, as opposed to expensive methods collecting a lot of human annotated data. Such approaches can be used to build a state of art perceptual model for *any* sound of interest described using a few signal processing features. This would not been achieved by directly mapping to the sound to latent space of the images (like SoundNet) due to data imbalancing issues, and relative short duration of such sounds. Daisy chaining high precision sound event detectors using signal processing combined with neural architectures and high dimensional clustering of unlabelled data is a vastly powerful idea, and can be explored in a variety of ways in the future.[2]

[1]Artificial Intelligence Laboratory, Department of Computer Science, Stanford University, Stanford, USA. Correspondence to: Prateek Verma <prateekv@stanford.edu>.

*Proceedings of the 37th International Conference on Machine Learning*, Vienna, Austria, PMLR 119, 2020. Copyright 2020 by the author(s).

[2]This research was supported in part by the Toyota Research Institute and in part by Amazon Robotics. The authors are also thankful to a computing grant from Google Inc.

## 1. Introduction

Audio scene understanding has been a subject that has been studied in depth for the past couple of decades (Bregman, 1994). In addition to making computers understand human speech, giving them the capabilities to hear and understand everyday sound has enormous applications. Giving computers ability to hear, see and speak on par with humans has been a long standing vision of early works in artificial intelligence (Papert, 1966). With the advent of convolutional models early success in vision-based CNN architectures has been translated to audio based research with proportionate gains in performance (Hershey et al., 2017). Various architectures from vision and natural processing have been successfully deployed to solve various problems in audio transformations (Haque et al., 2018), style transfer (Verma & Smith, 2018), speech recognition (Chan et al., 2015), learning latent representations (Haque et al., 2019), speech to speech translation (Guo et al., 2019), synthesis (Wang et al., 2017), perception (Verma & Berger, 2019). One application of learning good perception model has been in robotics in problems such as insertion and grasping. The work by (Lee et al., 2019) used shared latent representation across modalities such as vision, touch to solve the problem of interest. In the future, one can imagine that these latent-based representations will be universal irrespective of the domain to solve a particular task. For robotics, these have already been shown to advance the state of the art in problems with supervised/unsupervised approaches. Our work explores how we can augment similar auditory based representations. Additionally, we also propose to have these models learn from actual vast amounts of unlabelled data e.g. YouTube to solve a particular perception task in our case. The addition of sound in robotics application has been relatively untapped and may likely augment existing vision and touch based sensors in the future given similarities in learning algorithms and architectures in these domains (Lee et al., 2019). This work builds one such sub-block, namely can we build a perceptual model for any sound of interest for robotic application. Such models have similar architecture as existing one proposed in (Lee et al., 2019) and will help in learning latent representations for a particular scenario of interest e.g. scratching, rubbing, tapping. There have been works which build perceptual models for sound of interest

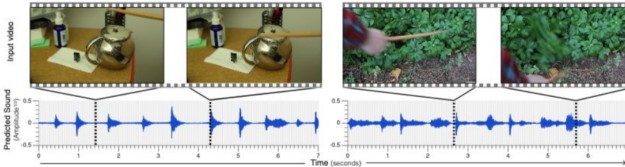

*Figure 1.* Diagram from (Owens et al., 2016) indicating creation of a dataset to record sounds while recording video of impact sound caused by beating a stick to different objects. This approach is not scalable to collect a lot of diverse types of data, and is time consuming.

by collecting vast amounts of data (Hershey et al., 2017). The work done by (Owens et al., 2016) tapped a variety of objects to predict the sound and material properties using touch sound as shown in Figure 1. However, such data collection is expensive as it uses a lot of resources, and cannot be scaled for a wider context and interactions. In this work, we propose a method of unsupervised learning a perceptual model for signals of interest, viz. touch sounds by daisy chaining signal processing to get sounds of interest with a high precision. We then use convolutional architectures to learn how to project sounds to a uniform manifold approximation space (McInnes et al., 2018) of the corresponding images. This bypasses the need of any labelled data.

## 2. Data-set

We took as input 5000 YouTube videos from a variety of real world scenarios. A subset of AudioSet (Gemmeke et al., 2017) corpus was chosen, which consisted of sounds like chop, tap, rub, slap, hammer etc. Due to building a high precision detector, we had a total of 3000 training examples, with some chosen for validation and testing. Given the availability of such datasets like YouTube-8M (Abu-El-Haija et al., 2016) and AudioSet, in the future, this can be scaled across a large variety of scenarios. We do not work with the availability of pretrained embeddings, as they are only trained for a particular application, and are not trained to infer fine-grained distinctions, i.e. in our case various types of sounds of touch.

## 3. Methodology

This section explain the ideas and methods that we have discussed so far to build a perceptual model of interest that can extract latent represetation of the sound of interest, and can successfully map the contents of the audio signal to a semantically meaningful space.

### 3.1. Signal Processing to Understand Sounds

A rich literature of hand crafted features was developed to capture various time and frequency domain characteristics

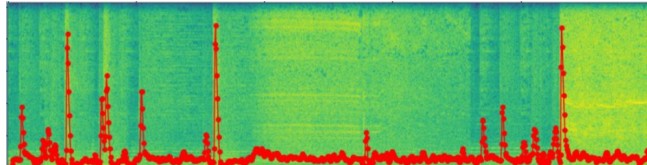

*Figure 2.* Spectral onsets computed for each of the time points in the spectogram. They have a strong correlation with impact and touch sounds

before the advent of deep learning (Peters, 2003 (accessed February 8, 2020)). We explain a few of these, in order to better motivate and emphasize their use. Some of the features that have been used both at a micro and macro level are zero crossings, spectral centroid, spectral flatness, onset strength, energy contour. Figure 2 explains a method of computing onset strength at each of the points in log-magnitude spectogram with the height of the red curve depicting the strength of the onset. We will explore a few of these features as described in the next section.

### 3.2. Detecting Sounds with High Precision

From the rich variety of features available to us, we explore how to characterize touch/impact sounds for the current work. Consider a time domain signal $x[k]$ and its corresponding spectrogram denoted as $X[k, m]$ where $k$ denotes the $k^{th}$ frequency bin computed at time instance $m$ typically every 10ms. We denote the following features as follows: Energy contour, $E[n]$ is defined as,

$$E[n] = \sum_{k=1}^{N} |X[k, n]| \quad (1)$$

Spectral centroid, as the name suggests is the center of mass of each of the spectral slices $X[:, n]$ and computed as the weighted average of the spectral weights over the spectral bin indices. Figure 3 describes the spectral centroid and the energy onsets of a setting where some utensils were moved and dropped. Spectral flatness $SF[n]$, is defined as,

$$SF[n] = \frac{\sqrt[n]{\prod_{k=1}^{N} |X[k, n]|}}{\frac{1}{N} \sum_{k=1}^{N} |X[k, n]|} \quad (2)$$

Spectral flatness is a measure of the impulsiveness of the sound or how the spectral spread of the spectral slice is across the frequency bins. Additionally, we also describe the spectral attributes or changes in terms of onsets.

A onset function $O[n]$, computed every 10ms is,

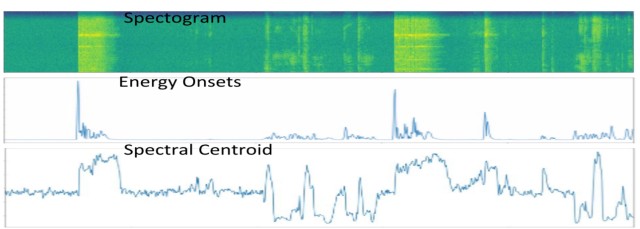
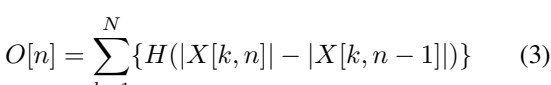

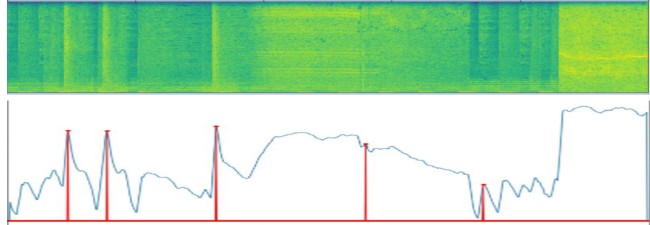

*Figure 3.* Example of a spectogram (top), energy onsets (middle) and spectral centroid of the spectogram slice (bottom)

*Figure 4.* Original Spectrogram (top) along with the energy contour and peaks chosen in red for training our system.

$$O[n] = \sum_{k=1}^{N} \{H(|X[k,n]| - |X[k, n-1]|)\} \quad (3)$$

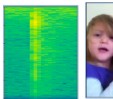 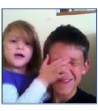  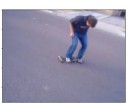 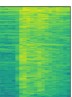 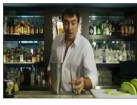

where the half wave rectification function is defined as $H(x) = \frac{|x|+x}{2}$. Another attribute, the number of zero crossings is defined as the number of times signal crosses the mean value in the waveform domain, quite often 0. Periodic signals will have a small average zero crossings in a frame whereas noisy/impulsive signals will have larger values. The touch sounds can be characterized using the following observations i) they consist of relative peaks in the energy envelope irrespective of the background noise present if any, ii) they usually are a wide-band event having a wide spectral spread as they are often associated with impulsive events iii) they exhibit peaks in spectral onsets. A smoothed energy contour is computed, and in addition to peaks, we also compare the depth of the peaks to correlate it with the impact/touch sound. In addition, we search for peaks in spectral flatness above a threshold and relative onset strength. We choose to build the models with a high precision and relatively poor recall. The Figure 4 describes a energy contour with the choice of the points chosen in red, in building a high precision detector.

### 3.3. Building Deep Learning Models

There has been a rich variety of work which uses convolutional and recurrent models to build audio understanding systems (Hershey et al., 2017) (Verma et al., 2019). Sound-Net (Aytar et al., 2016) leveraged a state of the art image understanding system to understand the contents of the audio signal, with a mean squared error loss to train a convolutional net to map wave-forms to latent space of images. However, such approaches do not work for the current problem of interest in building perceptual models for touch/impact sounds. These sounds are often too small, roughly around 40-100ms. Trying to map these sounds to the latent spaces, often ends up learning other sounds than the touch sounds due to data imbalancing problems. Given a spectra of say 10sec, such sounds will only account for less than 1% of the frames and the minimization criteria will only optimize the errors present in other 99% of the

*Figure 5.* Example of the sounds and the corresponding images. The representation is a log magnitude linear spectrogram with 129 bins and the duration of 200ms. Notice how we achieve rich variety along with the images as opposed to (Owens et al., 2016)

data points most of the times. The data imbalance problem in machine learning is still an active area of research, and people often mitigate such issues with sub-sampling, loss functions. However, trying to build such models in unsupervised setting is difficult to do as we are not aware of the position of sampling. The pretrained latent space (e.g. ImageNet Embeddings) characterizes the salient contents of the image, and not necessarily the subtle characteristics present for distinguishing the sounds. For all the three of the cases in Figure 5, the image embeddings will give a salient weightage of humans/person but the corresponding sound describe three distinct events namely slapping, skateboarding and placing a glass on a table. We can argue that we can have additional constraints like incorporation of a multi-category loss (Aytar et al., 2016), but it would not help us in understanding the contents and its correspondence to the other images present in the dataset. Instead of focusing on having diverse embeddings as target and addition of metadata information, we focus purely on a approach based on clustering. This will also mitigate the need to predict the non-salient parts of the images, e.g. person, drinks in the background etc present in the latent code. (Arandjelovic & Zisserman, 2017) produced meaningful results in learning latent representations for both images and sounds . This paper explores yet another way to learn unsupervised latent representations for the problem of interest, in this case audio perception of touch sounds.

### 3.4. Uniform Manifold Approximation

Uniform manifold approximation and projection (McInnes et al., 2018) is currently a state of the art clustering and dimension reduction algorithm that uses graph theory, fuzzy

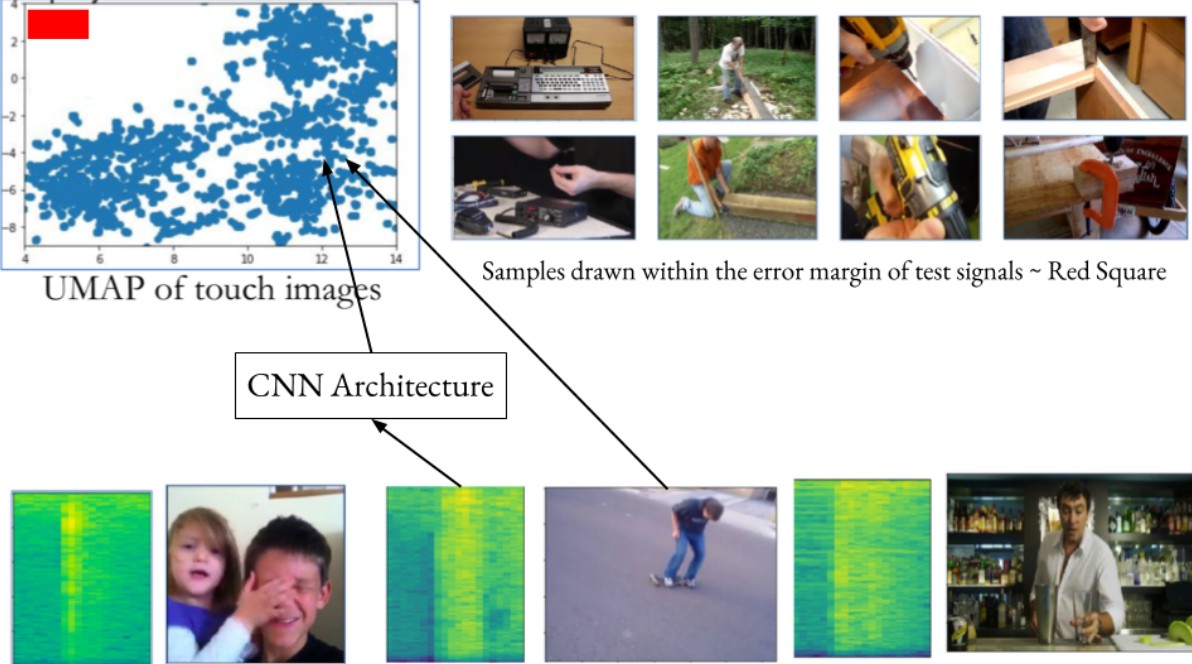

*Figure 6.* The basic architecture of the system. All of the images are clustered using unsupervised algorithm to map them to their UMAP space. A CNN architecture then learns to map the corresponding audio snippet, to their 2-D point in the UMAP space, which acts as a supervision to train the model. For mapping of a set of audio samples to their corresponding points in UMAP space, from a test set, the average error was depicted as the red square. Sampling from the red square show that the images form a semantically meaningful cluster often depicting the same category of sounds.

sets and topology. Across several datasets, it has been shown to retain the overall global structure present in the input signal of interest which is hard to capture with techniques such as PCA or TSNE (Maaten & Hinton, 2008). By adjusting two parameters namely number of neighbours as well as minimum distance, we can see we achieve much better separation and clustering of 3000 training images. UMAP embeddings also remove all the portions of the latent space (or embedding coordinates) that are not relevant to the problem of interest, and cluster images based upon its close similarity with other possible images present in the data-set. (Maaten & Hinton, 2008) on imagenet, showed how this clustering encodes the images into a space that is semantically meaningful. We compute UMAP representations for the latent codes learned from pretrained imagenet embeddings. In order to assess the quality of clustering, we sample images from UMAP space and its vicinity. We find that the images are clustered according to the type of impact seen in the images in addition to the overall scene. We see from Figure 6 that we can get drill sound, chiseling of wood, lab equipment based sound and dropping of a log of wood, within the margin of error of mapping test sounds as described in the next section.

## 3.5. Mapping sounds to UMAP space

In order to learn the latent representations of sounds, we use the position in the UMAP as a supervision. A 5 layer CNN model similar to (Arandjelovic & Zisserman, 2017) with 3x3 filters and 2x2 pooling was used, with Euclidean loss as the error criterion from the prediction points to the actual position in the UMAP space. The resolution of the spectogram was as follows: 10ms hop, 16kHz sampling rate, with 30ms window size and 512 pt FFT, with a total of 200ms of input in duration. This gives us a spectrogram of the size 257x20. The points in UMAP space will help guide us to the right cluster. Given a lot of data, it can help us perhaps in learning even finer grained distinctions present in the data. However, for most of the coarse settings such as "whether this was a wooden impact or not", we can already build state of the art models from access to a small number of such sounds. In order to evaluate the performance of our system, we hold out a portion of the test sounds, while *not* keeping the corresponding images out of the training set. The performance of the system is depicted in Fig 6, where the red square gives the average error in the x-y coordinates in projecting a test set of sounds into UMAP coordinates.

## 4. Conclusion

Mapping to UMAP space rather than labels or embeddings is a vastly untapped idea and can be used in several setups. We show how one can learn representations for the problem of interest with supervision from the position in clustered space. Additionally, combining signal processing with deep learning has been done to some extent in the past (Verma & Berger, 2019), but this work utilizes signal processing to collect cheap, high precision data from unlabelled sources. Such ideas can be used in the future to get any sound of interest and get corresponding modalities e.g. vision. This work also motivates, as to how when the signal of interest is small enough, how the traditional mappings to a latent representation would fail due to imbalance issues (between the number of points of signal of interest) while minimizing the euclidean distances between the latent representation and the embeddings learned by a convolutional architecture.

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
