# OpenReview forum: "Unsupervised Learning of Audio Perception: Learning to Project Data to UMAP space"
_ICML.cc/2020/Workshop/SAS — Submitted to SAS 2020_

### Official Review · AnonReviewer1 · 2020-06-28
**The title, abstract, motivation, related work, gap, experiments and results are unclear.**

**Rating:** 2
**Confidence:** 5

**Review:**

- It wasn't clear from the title or the abstract, what was this paper about. What is audio perception and what is being learned? What is the related work and gap? What are the contributions?
- "Audio perception is a key to solving a variety of problems?" How? There aren't citations about audio perception.
- Touch sounds is not a coined term, authors should define it and provide a list of all the classes used. Authors don't even say how many classes they used. Impact sounds (e.g. hitting and tapping) is a different group from Scraping sounds (e.g. rubbing, sawing)
-  "Such approaches can be used to build a state of art perceptual model for any sound?" What is a perceptual model? Any sound? This is really a stretch of scope considering the actual state of the field of Sound Understanding.
- The introduction starts talking about "Audio scene understanding", which is a different focus to sound (event) understanding, the actual topic of the paper.
- What is the motivation of the paper, is it to  explore what has been done before on a different field or to leverage from images as a mean of unsupervised labeling of sound events? "Our  work explores how we can augment similar auditory based representations",  "In this work, we propose a method of unsupervised learning a perceptual model for signals of interest"
- Section 3.2 starts right away choosing low-level features, and justifies them afterwards, I suggest to reverse the order. Moreover, the motivations are mostly reasonable, but lack justification with any literature reference.
- This is not exclusive of impact sounds " i) they consist of relative peaks in the energy envelope irrespective of the background noise present if any"
- No justification and evidence of the benefit of using low-level features over the sound event labels to select the initial set of sounds.
- No evidence for the justification: "Trying to map these sounds [impact sounds]  to the latent spaces, often ends up learning other sounds than the touch sounds due to data imbalancing problems",  it could instead be because of their duration or frequency range.
- No quantitative experiments, no diagram or clear explanation of the system, and there is an assumption that the reader understands UMAP.
- The conclusions are limited, here is an example: "Given a lot of data, it can help us  perhaps in learning even finer grained distinctions present in the data."
- References to the audio are lacking in the topics of DSP, psychoacoustics and sound event labeling.

---

> ### Author Response · Authors · 2020-07-09
> **Comments from Prateek about Anon Reviewer 1**
>
> Here it is in plain simple BBC English so that it can understood, if it was not clear from abstract, title and the paper. It is rather unfortunate and concerning. It seems that the reviewer has not understood it.
>
> 1. We want to build a model in order to distinguish different touch sounds. (Or any other category of sound) e.g. if I am hitting a wood or metal object.
> 2. We do not want to go around hitting objects, although it sounds fun, like what folks from MIT were doing. It is a straightforward problem then. Learn a conv net on sound to say what type of hit is.
> 3. Doing it in unsupervised way, approaches like sound-net do not work. Why? Touch sound occur for 20-50ms. If you do not have any labels, then mapping sound to latent space of the image from videos will learn all other kinds of ambient sound. The data is heavily imbalanced towards other sounds to learn such things. In a 10s video, there might be only 1-2 touch sound, and 99% of the other data would be other ambient sound.
>
> This is the whole reason AI algorithms are biased.
>
> Our solution:
> 1. Use signal processing, or feature to collect data cheaply with a "high precision" for the sound of interest. If we could have a high recall too, then the problem is solved. Signal processing alone cannot solve the problem.
> 2. Use embedding like T-SNE/UMAP/PCA to group together latent representations of images. The choice can be any clustered space. Why? Because it sees all the points in the data to group things together.
> 3. Map sounds to this clustered space.
>
> Lo and behold, without any annotations, one can now say if I was hitting something with metal or wood. How do we know this ?
>
> Take new sounds from the test set, and see where the sound maps to in latent space. Retreive the image from the database. How do we know if works? Sampling from error the test sound gives similar images to the input sound.
>
>
> ~ Prateek

---

### Official Review · AnonReviewer3 · 2020-06-29
**Solid background but narrow novel contributions**

**Confidence:** 4
**Rating:** 5

**Review:**

This manuscript describes an approach to learning unsupervised representations of audio.  The central approach is to learn a space using Uniform Manifold Approximation and Projection (UMAP) and then use this space to describe pseudolabels for use in training.

Strengths:
* Learning a cluster based representation using a different set of features is an interesting, if not entirely novel, approach.
* There is some evidence that this works.

Weaknesses:
* Much more time is spent describing other knowledge driven approaches to identifying sounds than is spend describing UMAP.  The rest of the approach is interesting only insofar as UMAP is trusted as a mechanism to construct an informative or useful feature representation.  I would recommend devoting more attention to this algorithm and assessing its value as a target for this task.  Consider the question, for example, why is the UMAP space preferable to a TSNE space, or even using K-Means centroids as a set of basis vectors to define a space?  The UMAP space is critical to this technique, but not compared against any alternatives.
* The evaluation is impressionistic. “The performance of the system is depicted in Fig 6, where the red square gives the average error in the x-y coordinates in projecting a test set of sounds into UMAP coordinates.”  Some testable measure along with the examples would be more convincing.
* “In order to evaluate the performance of our system, we hold out a portion of the test sounds, while not keeping the corresponding images out of the training set.”  What would happen if the images were removed from the training set?
* The spectrograms are treated as fixed sized data rather than sequences with potentially distinct sizes.  Why was that decision made? Is there a mechanism to extend this approach to treat speech as a temporal sequence rather than a static view.

Minor Presentation Comments:
* “Consider a time domain signal x[k] and its corresponding spectogram denoted as X[k, m] where k denotes the k th frequency bin computed at time instance m typically every 10ms.”  Overloading the subscript ‘k’ here is odd.  Consider using a different subscript for the time domain x[k] and the frequency in the spectrogram X[k, m]
* The introduction does not make clear what the central approach to the problem is.  It takes reading much of the paper to have a solid understanding of how the pieces described fit together, including understanding what is background, related work and what is actually being used in the described approach.

---

### Official Review · AnonReviewer2 · 2020-07-01
**Good Idea. Poor evaluation. No Baseline.**

**Confidence:** 3
**Rating:** 4

**Review:**

The paper claims to build a perception of touch sounds by mapping sound to UMAP space using some suggested features. The paper lacks rigor in experiments and evaluation about the effectiveness of the method (or using the specific suggested features).

---

> ### Author Response · Authors · 2020-07-06
> **Comment about Review from Authors**
>
> I agree that the experiments and evaluation lack rigor. The choice was made for a workshop submission vs a regular submission. We should have thrown in some numbers probably. We use suggested features only to collected data cheaply without any ground truth annotations. The specific suggested features were never used for training perception model. ~ Prateek

---

### Decision · Program_Chairs · 2020-07-01

**Decision:**

Reject

**Comment:**

Dear author(s),

Thank you very much for your submission at the ICML2020@SaS workshop (https://icml-sas.gitlab.io/). Based on the scores assigned by the reviewers, we regret to inform you that the paper was rejected. We got 26 submissions and we were only able to accept 13 papers. We invite you anyway to consider the feedback of the reviewers and to follow our upcoming workshop on July 17.